# Russian Citizens’ Attitude toward Insurance Policies as a Factor of Individual Economic Security

**DOI:** 10.3390/bs10010023

**Published:** 2019-12-31

**Authors:** Olga Medyanik, Olga Deyneka

**Affiliations:** Department of Political Psychology, St. Petersburg state University, Nab. Makarova, H. 6, 199034 Saint-Petersburg, Russia; o.deyneka@spbu.ru

**Keywords:** insurance behavior, individual economic security, policyholders

## Abstract

Today, insurance enables the functioning of the market system. In modern Russia, such a mechanism of protection against internal and external threats exists to ensure the economic security of each citizen. Indeed, individual insurance policies continue to gain importance as the most effective risk management tool to guarantee the safety of the health and property of private citizens. The goal of this study was to investigate Russian citizens’ attitudes towards insurance policies and investment bearing in mind the concept of personal economic security. Preparations for this study were focused on theoretical understandings of economic security problems, taking the field of insurance as an example. Our research consisted of four stages, with a total of 1794 participants. The results of this study can offer insight to improve the functioning of the insurance market in accordance with the framework of the Insurance Industry Development Strategy for the Russian Federation—2020. The obtained results can be used from both a political and economic standpoint in the development of a set of measures dealing with the control of financial institutions, promotion of financial literacy, preparation of courses for universities, and trainings for participants in the insurance market. Policyholders can also use this information to advocate for improved insurance programs for citizens.

## 1. Introduction

Today, in Russia, the development of the state insurance policy has entered a period of relative stability. However, two problems remain: (1) The lack of an insurance and investment culture among private citizens due to the widespread paucity of economic knowledge and financial literacy and (2) the need to consider state institutions in the adoption and implementation of legislation regarding insurance [1,2].

Because the insurance market is still largely controlled by state economic policy, we believe it is timely and necessary to conduct research with respect to political and economic attitudes on this issue and to address the following research challenges: First, there is a lack of research examining the relationship between citizens and insurance and investment policies. The information garnered from such research would facilitate the development of insurance programs that incorporate the trust and preferences of citizens with respect to different insurance products and agents in the insurance market. Second, there is a lack of scientific research on behavioral patterns related to insurance, which would provide insight into the economic security factors that affect the behavior of individuals with respect to insurance.

In light of this, from 2008 to 2019, we conducted a study of citizens’ attitudes toward insurance and investment policies as a factor of individual economic security. The study consisted of four stages. The first was a preparatory stage (2008–2016), during which characteristics describing attitudes toward insurance policy were collected. Surveys of1254clients of insurance companies in the St. Petersburg and Leningrad region were carried out via longitudinal research, using the observation method and semi-structured individual interviews. The most important determinants influencing the attitudes of citizens to insurance policies were defined, including the preferences of citizens (e.g., expediency of insurance, rational choice of insurance products), perception of risk, financial anxiety, and attitude to economic policies in general [3].

Inthe second stage of the pilot study (2017–2018), preferences with respect to insurance (PIP as developed by Deyneka and Medyanik) and the financial anxiety (FA as developed by Medyanik) of 108 clients of insurance companies from the St. Petersburg and Leningrad region were measured with the help of multifactor surveys, which we developed. 

In the third stage of the study(2018), the characteristics of 128citizens’ attitudes toward the economic policies of the Russian Federation and, in particular, toward insurance and investment policies, taxation, benefits policies, and general economic attitude (survey of attitudes toward economic policy (AEP) as developed by Deyneka and Medyanik), as well as economic attitudes survey(EA)developed by Deyneka and Zabelina), the Tolerance of Ambiguity Scale (TAS) by S. Budner in adaptation of G. U. Soldatova (2008), and the Satisfaction with Life Scale (SWLS) by E. Diner were measured [4,5].

In the fourth stage of the study (2019), we conducted our main research on citizens’ attitudes towards insurance policies (survey of attitudes toward insurance policy (AIP) as developed by Medyanik). The study involved 304 citizens (actual and potential policyholders) of the Russian Federation.

The design of the present study was based on Dr. O.S. Deyneka’s empirical work, which studied economic policy (the government) in different periods of Russian society from the perspective of different social groups [4,5]. In our study, we combined surveys and the method of longitudinal observation with semi-structured interviews. Including the observation method in the study made it possible to make the analysis more accurate by studying certain psychological characteristics of the subjects over a period of time.

The research scheme was based on the traditions of political and economic psychology developed at St. Petersburg State University and the Institute of Psychology of the Russian Academy of Sciences. This study took into account the scientific contribution of many authors studying the determinants of individual financial decisions, for example, when making decisions about savings [6,7,8], about preferences for compensation [8], and about insurance [7,9].The categorical apparatus of the study included the fundamental psychological concept of “psychological attitude”, as well as the concept of “individual economic security and financial well-being” [10,11,12,13,14,15,16,17,18,19], “subjective economic well-being” [6,18], “decision-making in the real world” [16,17,18,19], “insurance culture and economic confidence” [1,2,7,8,9], financial literacy, financial planning [7,16], economic conditions, economic values, economic beliefs in the context of insurance socialization [1,2,7,8,9], the fear of loss of life, health and property, financial anxiety, and financial ambiguity [18].

## 2. Materials and Methods

The longitudinal study (Stage 1) lasted for eight years from 2008 to 2016 and included the observation method in combination with the interview method developed during the first (preparatory) stage of the research. In total, the study sample included clients of different insurance companies in the St. Petersburg and Leningrad region (RESO, Ingosstrakh, ERGO RUS). There were 1254 people in the study group, who annually extend their previously concluded contracts of insurance.

The use of a qualitative method of analysis of insurance behavior over a long period of time was used to prepare for the quantitative methods of analysis.

The method of longitudinal research made it possible to obtain data on the process of socialization of policy holders and stable psychological characteristics of behavior. Rationality, risk, financial anxiety, attitude to insurance policies, and insurance trust were the criteria for the classification of types of insurance behavior. Further, the method assumed descriptions of insurers by types, characterized by relatively homogeneous characteristics determined by socioeconomic and psychological characteristics of their behavior in the insurance market. Each type of insurance behavior included expressed signs or a set of symptoms, which were developed and numbered for further interpretation of the results.

The study assessed the attitudes of citizens toward insurance as a financial institution in general and legislative initiatives in this area of economic policy. The measurement of loyalty to an insurance company and its proposed insurance products revealed both positive and negative views of the insurance industry.

Stott’s observation map [3] was used to construct the map of observation of the behavior of policyholders. The monitoring chart for insurance behavior included five symptoms of insurance behavior.

In the second stage (Stage 2), to improve the reliability of the data obtained by the method of longitudinal observation, methodological and empirical research related to the development and testing of quantitative methods for the study of the psychological and psycho-political characteristics of the relationship of citizens to insurance policies was developed.

To address the aims of the pilot study in accordance with the conceptual scheme of the study, multifactor surveys were developed (“preferences of citizens in insurance policy”(PIP) and “financial anxiety”(FA). Some of the statements from these surveys were then included in the fourth stage of the main study.

The design of the study, which was developed at the second stage, was based on the results of a longitudinal study of the behavior of policyholders and was aimed to identify preferences in insurance policy (expediency and benefit of insurance) and susceptibility to risk (acceptance, transfer, risk avoidance). The surveys were posted on the Survey Monkey online survey platform (https://ru.surveymonkey.com).

The first survey “preferences of citizens in insurance policy” (PIP), was designed to study the marketing preferences of respondents with respect to insurance The contents of the questionnaire included the following political–psychological phenomena: The request of citizens for state control over insurance market participants; principles of interaction between citizens and agents of the insurance market (banks, insurance companies); the ratio of citizens to insurance institutions; priorities in insurance policies; and the rationality of choice of insurance products, attitude to risk, and responsibility in the insurance sector. The study design was justified by the need to confirm two criteria: (1) Rationality (rationality of insurance (benefits); rationality of choice (price, availability, service)) and (2) risk (risk susceptibility, acceptance, transfer, risk avoidance). A total of 55 clients of different insurance companies in the St. Petersburg and Leningrad region (RESO, ERGORUS, Ingosstrakh), including 32 women and 23 men aged 25 to 45 years with an average to high income level, took part in the survey of citizens’ preferences with respect to insurance policies.

The second survey, named “financial anxiety”(FA) was filled with statements that reflect psychosomatic personality disorders that characterize financial anxiety. Insurance of risks by a citizen is a basic cause of concern and anxiety for personal economic security. Some common signs of financial anxiety include statements concerning feelings of depression or anxiety about finances, worries regarding embezzlement and loss of funds, the need to be frugal, discomfort and ambivalence in relation to the growth of wealth, the problems of transferring financial risks to the business of intermediaries, and inability to change the financial behavior of the volatility of the family budget. The statements were selected taking into account the accentuated mental forms of behavior associated with the problems that a person may face in economic life. 

This survey was designed to study the psychological and clinical aspects of the financial anxiety of consumers of the insurance market and other manifestations of economic life. The foreign community of experts in psychology and psychiatry identifies financial anxiety as part of a generalized anxiety disorder but has not yet recognized it as an official diagnosis. A total of 53 clients of different insurance companies in the St. Petersburg and Leningrad region took part in the survey on financial anxiety; there were 26 women and 27 men aged mainly from 25 to 45 years with an average to high income.

The purpose of the third stage (Stage 3) of the study was to study the psychological and political background of the attitudes of Russians to economic policies in the context of insurance (AEP), as well as tax, social policy benefits, attitude to charity, and quality of economic life. The rationale for the choice of the empirical base of the study was the analysis of both domestic and foreign studies on the development of problems of citizens’ attitudes to economic policies and economic institutions, such as insurance, investment, taxation, etc. [15,17,18].

Our study of attitudes toward economic policy and reforms using the example of the insurance industry was conducted in 2018, when citizens were most acutely aware of the aggravation of foreign policy and the pressure of sanctions from Western countries.

The third stage included the following set of psycho-diagnostic instrumentation: (1) A scaled, multifactor questionnaire regarding citizens’ attitudes toward economic policy (in the insurance context)—the AEP questionnaire (2018) by Deyneka and Medyanik [19]; (2) a survey of economic attitudes(EA)s by Deyneka and Zabelina (2018) [4,5]; (3)the Tolerance of Ambiguity Scale (TAS) by S. Budner in adaptation of G. U. Soldatova (2008) [3]; and (4) the Satisfaction with Life Scale (SWLS) by E. Diner [20].

The AEP questionnaire included three sets of statements, assessing attitudes toward (1) insurance and investment; (2) taxes and benefits; and (3) quality of life. The first block of the questionnaire was used to study respondents’ confidence in economic policy in general and in the insurance system in particular. For example, respondents were asked to assess their degree of agreement with the statement that the state, as a guarantor, needs to strictly control financial institutions (insurance companies, banks, pension funds). The questionnaire also evaluated insurance preferences (choice of compulsory and voluntary types of insurance, insurance planning, abandonment of insurance activities) and estimated confidence in market agents.

With the help of the second block of the questionnaire, attitudes toward tax and social policies were studied. Respondents were asked to assess their degree of confidence in the modern tax system. Some of the cited allegations concerned corruption in components of the system, for example, “Taxes—a feeding trough for officials”. Social policy was presented via statements concerning the issue of benefits for the poor and so forth.

The third block of the questionnaire was devoted to the subjective assessment of the quality of life and economic and psychological adaptation of the particular respondent.

The questionnaire assessed attitudes toward some internal political problems in the country and the financial management of the citizen.

The study involved 128 citizens (actual and potential policyholders) of the Russian Federation (including 87 citizens of St. Petersburg and 41 from other regions of the Russian Federation). The sample included 82 women and 46 men aged 18 to 68 years, i.e., it was heterogeneous. Respondents were represented by consumers of insurance services and citizens not related to insurance activities.

In the fourth stage (Stage 4), of the study (2019), we developed amoretto measure citizens’ attitudes of insurance policy (the AIP questionnaire), which was based on previous studies and certain aspects of theoretical analysis and longitudinal study and was intended for practical use. 

Based on our analysis of the theoretical concepts and our empirical research, the AIP questionnaire included the following five main themes:

Expediency of insurance (benefits). Citizens’ perceptions of insurance policy are based on their assessment of the feasibility and necessity of purchasing insurance products. The assessment of expediency is determined by the list of assumed risks, taking into account the probability of their occurrence and the awareness of losses that can occur under adverse circumstances.Rationality of choice (price, availability, service). Rationality largely depends on the marketing preferences of the consumer, which is determined by the benefits of purchasing an insurance policy, economic patriotism, and the financial literacy (education) of the citizen.Risk susceptibility (acceptance, transfer, risk avoidance). This indicator is determined by the individual citizen’s susceptibility to risk, especially with respect to whom the citizen is willing to transfer the responsibility for the preservation of his property in the case of danger: The state, insurance companies, relatives and friends, or himself.Financial anxiety (internal feeling of economic security, financial uncertainty). Characteristics of financial anxiety become a component of general citizens’ perception of insurance. The AIP model also reflects the positivity and negativity of the individual’s thoughts and expectations about his financial future, taking into account the manifestation of anxiety and anxiety for his stability and the experience of satisfying consumption in the insurance sector.Perception of insurance policy (insurance trust, psychological background, conditions, settings). This is based on the general trust of citizens, not only with respect to insurance institutions, but also to the economy in general. This indicator reflects attitudes toward the state as a guarantor of stability in the economy and a regulator of economic institutions. 

The study involved 304 citizens (actual and potential policyholders) of the Russian Federation. The sample was balanced by gender: 45 men and 55 women. By age, the survey participants can be divided into several groups: 23.2% (18–29 years), 29.9% (30–30 years), 30.2% (40–49 years), 11.7% (50–59 years), and 5% (60 years and older). The main occupation at the time of the survey was students (16.3%), public sector workers (19%), working in the commercial sector 54.3%, unemployed (3.3%), housewives (2.7%), and pensioners (4.3%).

## 3. Results and Discussion

The results of our study of the attitudes of Russian citizens toward state insurance and investment show a contradictory attitude of citizens toward insurance policies and a lack of psychological readiness to shift economic risks to insurance companies.

The term “economic security” is defined as a long-term state of personal financial stability, including provisions for the manifestation of external and internal threats to personal financial sovereignty and independence and the capacity for economic development, which provides the opportunity to meet and expand economic needs.

Longitudinal monitoring of the behavior of consumers of insurance services allowed us to identify four types of policyholders: Rational, anxious, trustful, and risky. We also identified four types of insurance behavior strategies: Rational and alarming, rational-risky, anxious-trusting, and risky-trusting.

The factors affecting preferences with respect to insurance policy, obtained from the sample of actual policyholders, include market makeup, paternalism affecting market activity, financial literacy, and constraints on insurance policies. Cluster analysis of empirical data confirmed that there are three types of policyholders: “Supporters of the market economy”, “paternalists”, and “neutrals”. “Paternalists” link the problems of the insurance industry and the distrust of insurance companies with poor state control of the activities of insurance companies and other financial institutions, as well as the lack of fair distribution of income of the population. For such citizens, the need to adopt fair economic policies is important. “Supporters of the market economy” distance themselves from the request for state control. The third category—”neutrals”—are those who clearly do not trust market institutions but who also lack faith in the ability of the state.

The factors identified in the financial anxiety questionnaire include factors affecting situational behavior that are associated with the loss of money, experience of debt, awareness of investment risks, need for insurance coverage, and financial optimism. Financial anxiety and uncertainty trigger consumer interest in insurance and investment products that can provide confidence in the future and prevent feelings of ruin or loss of property.

Our analysis of the gender differences in attitudes toward economic policy showed that women demonstrated higher social internality than men. They were more likely to attribute blame for economic problems to citizens and place more trust in the policies pursued by the state. Women, unlike men, advocated for targeted assistance in preferential security policies. They tended to believe that benefits should be received only by families in need. Based on our results, the women we sampled were more prone to stability, conservatism, trust, and “socially naiveté”; the men we sampled were more prone to risk and rational strategies with respect to economic behavior.

Our empirical research of the third stage shows the positive attitudes of citizens toward the insurance area factor of increasing economic security and tolerance to economic uncertainty: When citizens are more tolerant of uncertainty, they will find themselves more satisfied with the economic policy of their country. There is also a correlation between life satisfaction and attitude toward economic policies: The higher one’s satisfaction with life, the more positive one’s attitude toward the economic policies in Russia.

The results of this study, conducted on a sample of actual and potential policyholders, confirm the typological differences identified in our study of current policy holders. Our findings confirm that citizens generally fit into three categories with respect to their views on economic policies: “Paternalists”, “supporter of the market economy”, and “neutrals”.

Based on our theoretical analysis, we analyzed the insurance behavior of a citizen in a modern market economy, where the basis of perceptions of financial well-being is the economic security of the individual. A conceptual framework was proposed, modeling insurance behavior as a factor of the economic security of a person. The key elements that we identified may provide insight into the development of more effective insurance policies, taking into account the psychological uniqueness of modern Russia. 

Based on the identified factors affecting insurance behavior and our empirical study of attitudes toward insurance policies, we designed a model that favors the development of citizen trust in the insurance industry, considering issues of financial anxiety, the value of insurance coverage, financial literacy, the growth of an insurance culture, and economic security (in the context of insurance trust).

Factor analysis of the AIP data was used to validate our proposed model. We identified the five most important factors affecting trust in the insurance industry.

Thus, the model of attitudes toward insurance policies includes five factors that describe 41.25% of the obtained variance, empirically proving the correctness of the themes identified in the AIP model. Factor analysis helped to identify the structure of the survey responses, providing the basis for identifying the most important typologies of the respondents as follows: “Financial anxiety”, “accusing the authorities”, “supporters of the market economy”, “active policyholders”, and “paternalists”. Next, we considered the significant differences between these typologies and other variables in the AIP model. The analysis of the data showed that there are no significant differences between the choice of insurance products and the presence of financial anxiety in citizens who accuse the government of failing to implement sound economic policy (“accusing the authorities”). Significant differences were observed between the choice of insurance products and those groups that approve of features of market behavior (“supporters of the market economy”), the expediency of insurance (“active policyholders”), and the perception of risk (“paternalists”). The “supporters of market economy”, using rational strategies in economic behavior, were more likely to buy insurance policies, such as OSAGO (Liability Insurance), “KASCO” (Collision Damage Waiver Insurance), and real estate insurance (Table 1).

Significant differences were found in five risk perception factors (“paternalists”) between those who work for state-owned enterprises and those working in the commercial sphere. Among “paternalists”, more respondents tend to work in the public sector, and fewer tend to work in the commercial sector (Table 2).

There were significant differences in the three factors of market behavior, “Supporters of the market economy”, and the assessment of subjective income. Citizens who assess their income as low rarely use market strategies in their behavior; they are not characterized by rational attitudes. There are more middle-income citizens who adopt market policies (Table 3).

We also found differences in the levels of financial anxiety (“Financial anxiety”) between the poorest and the wealthier groups, such as those “who can buy, for example, basic household appliances”. 

Those experiencing financial anxiety and uncertainty are more likely to be those who are experiencing financial difficulties and whose families only have enough money for food. Respondents from families with average incomes and above showed less financial anxiety and concern for their future. There were significant differences in attitudes toward economic policies (“accusing the authorities”) with respect to the socioeconomic status of the family. Citizens whose families have average and above-average incomes are more likely to blame the government for the economic problems of the country and tend to be more dissatisfied with the consumer situation in the Russian Federation. The typologies we identified had a strong influence on the choice between compulsory or voluntary insurance programs (Table 4).

In order to predict the influence of these typologies on the choice between compulsory and voluntary insurance programs, an analysis was carried out using logistic regression—a method of constructing a linear classifier that allows us to estimate the probability of certain objects belonging to classes where the dependent variable takes a value of 0 if the respondent does not use voluntary insurance programs and 1 if the respondent does use voluntary insurance programs. In the calculation, the scores of 20 respondents who had no insurance experience at all were removed.

Compulsory programs include insurance programs in which the policyholder is forced to purchase an insurance policy at the request of banks or other structures, for example, mortgage insurance (the policyholder is forced to purchase a mortgage insurance policy at the request of the bank when obtaining a loan for an apartment or house), travel insurance for traveling abroad (the policyholder is forced to present such insurance when obtaining a visa), and mandatory insurance programs. All other programs (KASCO, real estate, life, accident, VHI) are considered voluntary types of insurance, which are purchased at the request of the insured.

As a result of this analysis, a final model was developed (Table 5).

The results of the regression analysis show that it is more likely that respondents categorized as” relevant insurers” and “supporters of the market economy” prefer to participate in involuntary insurance programs. Thus, an appositive attitude toward insurance policies is more likely among citizens who have already had experience with the insurance sector and citizens working in the commercial sector of the economy and use rational and market strategies in their economic behavior.

In summing up the main results of our analysis of the attitudes toward insurance policies, we can make the following conclusions.

Today, Russian citizens can be divided into the following categories based on their attitudes toward insurance policies:Paternalistic citizens are those who often work in the public sector, do not trust insurance institutions, and consider it impractical to transfer their property and non-property risks to private businesses, insurance companies, or other agents of the insurance market. They believe that all insurance contingencies should be the responsibility of the state and prefer to limit consumption by buying only compulsory insurance programs, such as OSAGO.The next group consists of citizens who work mainly in the commercial sector with average incomes. They approve of the market policy of the state and increase their consumption of insurance products, including voluntary types of insurance. They utilize rationality in making economic decisions.The most financially anxious citizens, who assess their income as low and do not participate in insurance activities, rarely use market strategies in their behavior; they are not characterized by rational attitudes.There is a group of citizens who accuse the government and politicians of failing to implement sound economic policies and who are dissatisfied with the options for consumers. Most often, these citizens have average and above-average incomes. However, these citizens are less likely to experience financial anxiety regarding their future.There is a category of citizens for whom insurance is the main strategy for the preservation of their assets. They acquire not only compulsory insurance programs but also voluntary ones.

## 4. Conclusions

Trust in the insurance industry is a necessary component of the investment behavior of consumers. Citizens not only require knowledge but also faith when making the serious, long-term decision to shift the responsibility for their financial security to insurance market agents.

It is necessary to improve the financial literacy of citizens with materialistic attitudes, to increase trust in the private insurance industry, and to develop rational strategies with respect to citizens’ economic behavior.

## Figures and Tables

**Table 1 behavsci-10-00023-t001:** Significant differences between groups using insurance products (N = 304).

	Mortgage Insurance		“OSAGO”		Travel Insurance		Real Estate Insurance		“KASKO”		Health Insurance		Accident Insurance		Life Insurance		Did not Apply to Insurance Companies	
	M	SD	M	SD	M	SD	M	SD	M	SD	M	SD	M	SD	M	SD	M	SD
“Supporters of the market economy”	0.075	0.133	0.065 *	0.069	0.036	0.080	0.343 ***	0.107	0.155 **	0.082	0.142	0.105	0.267	0.432	0.199	0.120	0.492	0.204
“Active policyholders”	0.150	0.119	0.044	0.072	0.051	0.083	0.322 ***	0.101	0.265 ***	0.081	0.337 ***	0.104	0.569	0.261	0.371 ***	0.105	−0.201	0.215
“Paternalists”	−0.214	0.123	−0.007	0.070	0.056	0.077	−0.232 *	0.106	−0.015 *	0.083	0.193 *	0.107	−0.448	0.276	−0.044	0.132	0.180	0.324

Note: Accuracy levels of differences are indicated by the number of stars-symbols: * *p* ≤ 0.05; ** *p* < 0.1; *** *p* < 0.01.

**Table 2 behavsci-10-00023-t002:** Significant differences between groups with different employment status (N = 304).

	Students		Work in State Enterprise		Work in Commercial Sphere	
	M	SD	M	SD	M	SD
“Paternalists”	−0.178	0.142	0.421	0.121	−0.165	0.078

**Table 3 behavsci-10-00023-t003:** Significant differences between groups with different estimates of subjective revenue (N = 304).

	Very low		low		middle		high		Very high	
	M	SD	M	SD	M	SD	M	SD	M	SD
“Supporters of the market economy”	−1.144	0.157	−0.437	0.137	0.125	0.069	−0.063	0.209	1.362	0.088

**Table 4 behavsci-10-00023-t004:** Significant differences between groups with different assessments of the socio-economic status of the family (N = 304).

	Enough for Food, but It Is Hard to Buy Clothes (1 Group)		Enough for Food and Clothes, but Buying Fridge or TV is Difficult (2 Group)		Able to Buy Appliances, but Should Get Loan to Buy a Car (3 Group)		Enough Money but not for an Apartment or House Purchase (4 Group)		No Financial Difficulties, Able to Buy an Apartment or House if Necessary (5 Group)	
	M	SD	M	SD	M	SD	M	SD	M	SD
“Financial anxious”	0.922	0.243	0.153	0.174	−0.013	0.078	−0.335	0.119	−0.622	0.292
“Accusing the authorities”	−0.130	0.247	0.163	0.175	0.184	0.080	−0.412	0.147	−0.033	0.469

**Table 5 behavsci-10-00023-t005:** Influence of the typology of the respondents on the choice of compulsory or voluntary insurance programs (N = 304).

	Coefficient	*p*-Value
Constant	1.44293	*p* < 0.001
“Financial anxiety”	−0.08752	
“Blaming authorities”	0.20463	
“Supporter of the market economy”	0.60533	*p* < 0.001
“Active policyholders”	0.94361	*p* < 0.001
“Paternalistic”	−0.21688	
Pseudo R-squared	0.26

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
