# Peer review of "Russian Citizens’ Attitude toward Insurance Policies as a Factor of Individual Economic Security"

_behavsci, 2019, doi:10.3390/bs10010023_

Round 1

Reviewer 1 Report

The paper deals with the attitude of the population of Russia towards the insurance policy of the country.

The keywords are selected in an inappropriate way. Keywords should be represented by nouns rather than by succession of a few words. For instance, "the typology of the policyholders" shouldbe substituted simply by "policyholder" or if you would like to emphasise the typology itself, then "policyholder typology". Besides this, the definite articles are not suitable for usage among keywords.

The literature review consists of only 4 references, which 3 of them are only references to the previous own publication from. It is absolutely unacceptable.
The Materials and Methods section involves information about 1,254 respondents, but a sum of 1793 respondents is mentioned in the Abstract. Why is it so? This section is quite rich, but it lacks a crucial information. The data were collected from a sum of the inhabitants, but there is no evidence about classification of the selected respondents. It is an important point in order to ensure a suitable quality of the obtained answers. In this section too, a sentence on lines 97 to 101 has no sense. Perhaps, it would be better to concentrate on quality of the text than on quantity. One paragraph expressing a proper aim is better than two paragraphs talking in a very vague way.

The first paragraph of the Results and Discussion section involves references by names in the brackets. It is unsuitable way how to refer to the other sources. The proper references should be included. Also, the reference on line 217 is placed in the middle of the sentence instead of the end of the sentence. The analysis outcome visualised in Table 1 is nowhere explaind from a methodological point of view. It looks like a regression model. But it is nowhere mentioned. Why? What type of coefficient it is in the second column? Also, no interpretation of this model is carried out in the text. This is another absolutely unacceptable point for the scientific paper.

The list of the references consist of only 6 references. It is a extremely low number. It should be several times higher number.

The text of the paper should be based on the particular English grammar – American or English. Now, it is a mixture of both ones.
The whole text is written in a strange way – word order is sometimes unusual for the English grammar. Also there are some other points that should be corrected – for instance:
– in the Abstract on line 21: "The obtained results can be used in politics and economics..." instead of "The obtained results can be used in Politics and Economics..." – there is no need to use the capital letters;
– in the Introduction on line 45: "from 2008-2019. " instead of "from 2008-2019. "
– in the Introduction on line 45: "Its..." instead of "Its’";
– in the Materials and Methods on line 78: "1,254" instead of "1 254";
– in the Results and Discussion section on line 323: "Nowadays, citizens..." instead of "Nowadays citizens...".

Author Response

Dear Reviewer!

Thank you very much for the review of our article. You made very important comments. We treid to fix them. The entire text was  a new translation into English by the editors of the magazine. I am sending you a new version of the article. Kind regards

Reviewer 2 Report

The paper needs comprehensive English editing. Many sentences are long and unclear (e.g., lines 58-61), and the choice of terms is awkward. For example, it is not clear what the authors mean when they discuss inventory. This makes the reading difficult and does not enable the readers to understand what the authors did.

Moreover, even when the language is clear, the description of experiments, sample and methodology is confusing. For example, it is not clear how 1,793 took part in the study (line 18) – I added up the numbers of participants in all stages and got a much smaller number. Later, the authors write that the sample includes 3,304 people.

An additional major problem is that the paper does not review at all prior literature on the determinants of individual financial decisions. There is ample literature on these topics (see, for example, Mugerman, Sade & Shayo 2014 on savings decisions, Abudy and Shust 2012 on compensation preferences). More specifically, numerous studies examine insurance decisions (e.g., Johnson et al. 1993). The authors need to add a literature review section, where they will review the literature and clarify what is their contribution.

Third, the logistic model whose results are reported in Table 1 is not presented in the paper, so it is not clear what this model controls for. Based on the table, it seems that the model does not control for personal characteristics affecting insurances decisions such as gender, age, education and wealth.

Minor comments:

Please present descriptive statistics of the sample.

Describe the variables in Table 1 and their possible values.

References

Abudy, M., & Shust, E. (2012). Employees’ attitudes toward equity-based compensation. Compensation & Benefits Review44(5), 246-253.‏

Johnson, E. J., Hershey, J., Meszaros, J., & Kunreuther, H. (1993). Framing, probability distortions, and insurance decisions. Journal of risk and uncertainty7(1), 35-51.‏

Mugerman, Y., Sade, O., & Shayo, M. (2014). Long term savings decisions: Financial reform, peer effects and ethnicity. Journal of Economic Behavior & Organization106, 235-253.‏

Author Response

Dear Reviewer!

Thenk you very much for the reviev of our work. You made very important comments.  We tried to fix them. The entire text was a new translation into English by the editor. I am sending you a new version of the article. Kind regards.

Round 2

Reviewer 1 Report

The amendments were done according to the required changes.

Author Response

Dear Reviewe!

Thanks for the next review of our article. In this version, we tried to correct inaccuracies in the English.

The word @empirical" was deleted (line 17). Specified the numder of respondents: altogether-1794 (Stage 1 - 1254 people (lines 45 and 89); Stage 2 - 108 people (line 53); Stage 3 - 128 people (lines 44 and 181); Stage 4 - 304 people (lines 210 and 64). Decrypted terms:"OSAGO"(MTPL Insurance), "KASCO" (Motor Hull Insurance).

Kind regards

Reviewer 2 Report

The current version of the paper shows great improvement. The paper is readable now and it enables the reader to understand the research the authors had carried out. Nevertheless, there are some issues that should be addressed:

First, the numbers of participants do not add up – how many people participate in the research altogether? 1,794 (line 17) or 1,254 (line 45 and line 87)? Were there 128 (line 55) or 127 (line 179) participants in the third stage? How the numbers of participants in all stages sum up to the total? The authors need to clarify this and make sure the numbers throughout the manuscript are consistent.

Second, the manuscript analyzes active policy-holding as an independent variable that explains attitudes towards insurance and insurance decisions. See, for example, the findings that "Active policyholders tend to buy expensive voluntary insurance programs" (lines 284-285). However, by definition, active policy-holders have a greater tendency to buy insurance than non policy holders. Therefore, the fact that a person is an active policy-holder is a result of all other personal characteristics, and not a characteristic by itself. Therefore, the authors should exclude this parameter as an explanatory variable, or econometrically address the endogeneity.

Lastly, the authors should explain what are OSAGO and KASCO insurance plans they refer to (e.g., line 284) and use in their empirical analysis.

Author Response

Dear Reviewe!

Thenks for the next reviev of our article. In this version,  we tried  to correct inaccuracies in the English.

1. The word “empirical” was deleted (line 17).
2. Specified the number of respondents: Stage 1 - 1254 people (line 45 and line 89); Stage 2 - 108 people (line 53); Stage 3 - 128 people (line 44 and line 181); Stage 4 - 304 people (line 64 and line 210).
3. Decrypted terms: "OSAGO"(MTPL Insurance), "KASCO" (Motor Hull Insurance).

4. Deleted data: "active policy-holders have a greater tendency to buy insurance than non policy holders" (line 285-286).

If we need to make more adjustments, we will be grateful if you point to them.

Best regards
